# Dietary Protein Intake Dynamics in Elderly Chinese from 1991 to 2018

**DOI:** 10.3390/nu13113806

**Published:** 2021-10-26

**Authors:** Yifei Ouyang, Tingyi Tan, Xiaoyun Song, Feifei Huang, Bing Zhang, Gangqiang Ding, Huijun Wang

**Affiliations:** 1National Institute for Nutrition and Health, Chinese Center for Disease Control and Prevention, Beijing 100050, China; ouyyf@ninh.chinacdc.cn (Y.O.); sxydljk@126.com (X.S.); huangff@ninh.chinacdc.cn (F.H.); zhangbing@chinacdc.cn (B.Z.); dinggq@chinacdc.cn (G.D.); 2School of Human Nutrition, McGill University, Montreal, QC H9X 3V9, Canada; tingyi.tan@mail.mcgill.ca

**Keywords:** dietary protein, trends, aging

## Abstract

Unique rapid urbanization-related changes in China may affect the dietary protein intake of the aging population. We aimed to evaluate trends in dietary protein intake and major food sources of protein and estimate conformity to the dietary reference intakes (DRIs) in the elderly Chinese population. A sample of 10,854 elderly adults aged 60 years or older, drawn from 10 waves of the China Health and Nutrition Survey (CHNS) between 1991 and 2018, was included. Protein intake data were obtained on the basis of 3-day, 24 h dietary recalls. The dietary protein intake among elderly Chinese individuals declined from 63.3 g/day to 57.8 g/day over the 28-year period, with a −0.032 ± 0.0001 g/day change per year (*p* < 0.05). There was a significant increase in the proportion of subjects with a protein intake level below the estimated averaged requirement (EAR) and a reduction in the proportion of subjects consuming protein above the recommended nutrient intake (RNI) across all population subgroups. Cereals ranked as the major sources of dietary protein, although their contribution to dietary protein gradually decreased as time went on. The contribution from meat steadily rose from 18.2% in 1991 to 28.7% in 2018. The proportion of energy gained from fat increased notably, reaching 34.2% in 2018. The elderly Chinese population experienced a significant reduction in dietary protein intake. Although the transformation of dietary patterns had positive effects on improving protein quality due to increases in animal source food, some elderly Chinese individuals currently face the risk of inadequate dietary protein intake.

## 1. Introduction

With the coming aging of society, the rapid growth of the elderly adult population has become a global issue. By 2030, one in six persons in the world will be 60 years of age or older [1]. China is also experiencing an appreciable increase in its elderly population [2] and has the largest population of older persons in the world [3]. In the 2020 Chinese population census, there were about 264 million people who were aged above 60 years, which accounted for 18.7% of the total population. Aging is one of the most important risk factors for functional decline, with serious health consequences including frailty, sarcopenia, cognitive dysfunction, and mortality [4].

An adequate intake of dietary protein in the aging population has been proposed as a potentially important modifiable factor not only for the maintenance of one’s body mass, muscle strength, energy balance, and physical performance, but also for counteracting inflammation, oxidation, and their downstream catabolic effects, thereby lowering the prevalence of age-related diseases [5,6,7,8,9]. The Chinese Nutrition Society established the current DRIs for protein in 2013, which included the EAR and the RNI. The EAR for protein is 60 g/day for male adults aged above 18 years and 50 g/day for female adults aged above 18 years (0.9 g/kg/day). The RNI for protein is 65 g/day for male adults aged above 18 years and 55 g/day for female adults aged above 18 years (1.0 g/kg/day). The EAR and RNI are defined as the minimum amount of dietary protein required for 50% of the population and for nearly the entire population, respectively. Previous studies showed a reduction trend in daily protein intake among Chinese adults over the past few decades [10,11]. Compared with younger individuals, older adults are less responsive to the anabolic stimulus of similar doses of amino acid intake and are often more vulnerable to losses in muscle mass and physical function [12]. However, there is no difference in the currently recommended daily amount of protein intake for young and older adults. A cross-sectional study showed that less than one-fifth of elderly adults (aged ≥ 60 years) had met the DRI recommendation for protein (<15% for total energy) [13]. Consequently, it is necessary to understand the dietary protein intake transition among elderly Chinese individuals to inform both future research and policy guidelines.

Besides quantity, the quality of the consumed protein is also essential for promoting muscle health. Researchers considered animal-based protein an excellent source of high-quality protein, as it contains a large quantity of essential amino acids (EAAs) [14]. Following the rapid economic growth and urbanization in China over the past few decades, the Chinese diet underwent rapid Westernization, characterized by increased consumption of animal source foods, processed foods, and sugar-sweetened beverages, whereas the consumption of coarse grains, legumes, vegetables, and fruits remains low [15,16]. Besides, eating away from home has become a large part of the Chinese diet. Changes in the economy and modernization can affect dietary protein intake at the population level. However, little is known about major food sources of protein, or the prevalence of protein intake below the EAR and above the RNI among elderly Chinese individuals. Therefore, the present study aimed to characterize dietary protein intake trends over the past 28 years (1991–2018) and assess trends in conformity with protein-specific DRIs in the elderly Chinese population, as well as estimate major food sources of protein and the energy contribution transition.

## 2. Materials and Methods

### 2.1. Data Source

We utilized the data from the CHNS, a longitudinal household-based study of the Chinese population. A multistage, stratified sampling design was used to ensure that the CHNS provided a fair representation of both urban and rural areas and minimized the selection bias. The CHNS aimed to capture the economic, sociological, and demographic transformations that occurred in China and to reveal the effects on the health and nutritional status of the Chinese population. Health workers were trained and certified to collect the data. Additional details regarding CHNS data have been reported in detail elsewhere [17].

Ten waves of the CHNS were included in this study, which were conducted in 1991, 1993, 1997, 2000, 2004, 2006, 2009, 2011, 2015, and 2018. We excluded disabled persons and those with missing data for diet, age, and gender. Finally, a total of 10,854 participants aged 60 and above with complete data collection were included in the analysis. The survey protocols, instruments, and processes used to obtain informed consent for this study were approved by the institutional review committees of the University of North Carolina at Chapel Hill, as well as by the National Institute for Nutrition and Health, which is affiliated with the Chinese Center for Disease Control and Prevention (201524). All participants provided informed written consent.

### 2.2. Dietary Intake Measurement

Detailed dietary intake information on three meals and snacks for each individual was obtained through face-to-face interviews on three consecutive 24 h dietary recalls, including two weekdays and one weekend day [18]. The trained staff used a tablet (Lenovo ThinkPad Tablet2, Lenovo, Beijing, China) to record information over the three consecutive days using food models and picture aids, in addition to food weighing.

### 2.3. Dietary Estimation and Food Sources of Protein

According to the China Food Composition Table, we computed the dietary protein intake (g), fat intake (g), and carbohydrate intake (g) from each food the participants consumed during the survey period. The total energy intake was estimated as the sum of energy from proteins, fats, and carbohydrates. The percentage of energy obtained from each nutrient was calculated as the proportion of energy from the nutrient of the overall energy intake. For example, for total protein, the percentage of energy = (total protein [g] × 4 [kcal/g])/total energy intake (kcal) × 100 [19].

Based on the Chinese DRIs, we used the EAR and the RNI cut-off points to estimate the dietary protein intake levels. The EAR for protein intake is 60 g/day for males and 50 g/day for females (0.9 g/kg/day). The RNI for protein intake is 65 g/day for male and 55 g/day for females (1.0 g/kg/day). Food items contributing to dietary protein intake were grouped according to the China Food Composition Table. We listed the top five food groups, including cereals (cereal and cereal products), meat (livestock, poultry, and aquatic products), legumes (dried legumes and legume products), vegetables and fruits, and milk and eggs. Other groups were not included since they only share a contribution of less than 3%, respectively.

### 2.4. Measurement of Sociodemographic Characteristics

Questionnaires were used to collect individuals’ information, including age, gender, education level, income level, and residential area. We grouped participants into two gender groups (male and female), two age groups (60–69 years and 70 years and above), two education levels (primary/illiterate and middle school and above), three income levels (low, middle, and high), and four residential areas (city, suburb, town, and village).

### 2.5. Statistical Analysis

We expressed continuous variables as medians (interquartile range), and categorical variables as numbers (percentages). Chi-square tests were used to compare differences among categorical variables. Trends in dietary protein intake were assessed by multivariable linear regression, treating the survey year as a continuous variable. Trends in conformity with protein-specific DRIs were tested using multivariable logistic regression models, including the survey year as a continuous variable. Per-year change was also calculated. Interactions between these variables and the survey year were examined by adding the product term of the time and variable of stratification into the multivariable linear regression model or the multivariable logistic regression model. The data were adjusted for all covariates, such as age, gender, education level, income level, and residence area. All data analyses were performed by SAS 9.4 (SAS Inc., Cary, NC, USA), and statistical significance was set at a two-tailed *p* < 0.05.

## 3. Results

### 3.1. Participant Characteristics

Table 1 describes the sociodemographic characteristics of the study samples in the CHNS wave. From 1991 to 2018, the proportion of participants aged 60–69 and above 70 years were around 60% and 40%, respectively. The proportion of male and female participants was 50% each. The proportion of participants with a primary/illiterate education level declined from 90.6% to 49.9%, whereas the percentage of individuals with middle school education and higher increased from 9.4% to 50.1%. The percentage of city residence increased from 27.4% to 40.5%.

### 3.2. Trends in Dietary Protein Intake Distribution

As shown in Table 2, from 1991 to 2018, the total dietary protein intake decreased among elderly Chinese individuals from 63.3 g/day to 57.8 g/day, with a −0.032 ± 0.0001 g/day change per year (*p* < 0.05). There were significant declines in dietary protein intake across all population subgroups (*p* < 0.05 for the trend for all). Older people who were aged 60–69 years showed a stronger decrease than those who were aged above 70 years. Having a higher education level and having a higher income decreased dietary protein intake more quickly compared to their respective counterparts. Additionally, participants living in cities showed a stronger decrease than participants living in villages.

### 3.3. Assessment of Dietary Protein Intake Level in the Study Participants

Table 3 and Table 4 show the trends in the proportion of daily protein intake below the EAR and above the RNI among the study participants. Overall, the prevalence of protein intake below the EAR increased from 35.6% in 1991 to 44.0% in 2018, whereas the prevalence of protein intake above the RNI declined from 56.7% in 1991 to 47.3% in 2018. There were significant increases in the proportion of subjects with a protein intake level below the EAR and reductions in the proportion of subjects consuming protein above the RNI across all population subgroups (*p* < 0.05 for the trend for all). Both the largest increase in the prevalence of protein intake below the EAR and the largest decrease in the prevalence of protein intake above the RNI were observed in participants living in villages. A higher prevalence of protein intake below the EAR and a lower prevalence of protein intake above the RNI were observed in participants aged above 70 years, in participants with only a primary/illiterate education level, and in those living in villages, compared to their counterparts.

### 3.4. Contribution Percentages of Food Sources to the Total Dietary Protein Intake

Figure 1 reveals the top five food sources, which shared more than 90% of the contribution to dietary protein intake over the 28-year study period. Cereals ranked as the major sources of dietary protein in the longitudinal study, although their contribution to dietary protein gradually decreased as time went on, reaching 33.6% in 2018. The contribution from meat steadily rose from 18.2% in 1991 to 28.7% in 2018. The percentage of the contribution from legumes decreased slightly from 10.8% in 1991 to 7.3% in 2018. Vegetables and fruits followed with little change. The contribution of milk and eggs showed an increasing trend, from 3.8% in 1991 to 8.0% in 2018.

### 3.5. Energy from Carbohydrates, Proteins, and Fats

Figure 2 presents the profile of the energy contribution transition among older Chinese people from 1991 to 2018. The energy from protein plateaued at 12% during the overall period. The proportion of energy provided from fat increased from 24.9% in 1991 to 34.2% in 2018, which exceeded the recommended 30%. Nevertheless, the energy from carbohydrate steadily dropped from 62.6% in 1991 to 52.9% in 2018.

## 4. Discussion

From 1991 to 2018, we observed a significant reduction in daily protein intakes in all categories of elderly adults in China. As a consequence, there was a significant increase in the proportion of subjects with a protein intake below the EAR and a decline for those with an intake above the RNI across all population subgroups. There are some elderly Chinese individuals currently facing the risk of inadequate dietary protein intake. The quality of dietary protein among elderly Chinese individuals improved as the percentage of the contribution from animal-based protein gradually increased.

China, with its significant elderly population, is one of the countries facing the most serious population aging-related challenges. Adequate protein intake is suggested for improving the physical performance and well-being of older adults. However, there is a decreasing trend in dietary protein intake among elderly Chinese people. The median protein intake among males (62.8 g/day) was already lower than the RNI (65 g/day) in 2018. Similarly, females with a 53.9 g/day intake were below the RNI (55 g/day). Hence, almost half of the elderly Chinese population did not meet the individual protein intake requirement, which may result in serious adverse consequences for healthy aging. Furthermore, epidemiological studies have suggested that current dietary protein intake recommendations may be insufficient for maintaining muscle mass and strength for the elderly and for improving their health [20,21]. The requirement for a larger dose of protein to generate responses in elderly adults similar to those in younger adults provides the support for a beneficial effect of increased protein intake in older populations [22]. In the current study, we found a substantial increase in the proportion of participants with a protein intake level below the EAR and a reduction in the proportion of participants consuming protein above the RNI in every subgroup evaluated over the past 28 years, particularly among those of older ages, those with primary education/illiteracy, and those living in villages. The reason may be that the cereal intake among village participants was much higher than that among other areas and cereals ranked as the major sources of dietary protein. The contribution from cereals was 56.2% in 1991. Similarly, a low protein intake is also very common in older adults in other Asian countries [23]. Due to widespread publicity regarding protein-related health benefits and the increased variety of protein-rich food products, the majority of the US population continued to meet or exceed both the EAR and the RDA, respectively [1,24]. Although dietary protein intake declined in the elderly Chinese population from 1991 to 2018, the percentages of energy gained from protein steadily remained between 11–13%. An explanation may be the chronological energy intake reduction among the elderly population [25].

Besides quantity, the quality of the consumed protein plays a critical role in the well-being of elderly adults [14]. The current analysis showed a sizable change in food sources of protein among elderly Chinese individuals. Although cereals continued to rank as the leading source of protein, the contribution percentage of cereals steadily decreased and gradually shifted toward meat products. From 1991 to 2018, cereal intake declined by 22.7%, whereas meat intake increased by 10.5%. This significantly reduced consumption of plant-based foods may be a potential reason for the decline in protein intake, which was higher than the increased consumption of animal-source foods. The present study showed that the contribution of animal protein and soy protein to the total dietary protein intake increased from 32.8% in 1991 to 44.1% in 2018, which was within the recommended range (30–50%). In general, animal-based foods are recognized as a superior source of protein because they have a complete composition of EAAs, with high digestibility (>90%) and bioavailability [20]. The composition of protein among elderly Chinese people has improved due to an increased intake of animal-based protein and soy protein.

Although the elevated intake of animal protein improves the quality of dietary protein, it is also associated with a risk of obesity, osteoporosis, high blood pressure, and other chronic diseases due to the high levels of saturated fatty acids and cholesterol in meats [26]. In the current analysis, we found that the percentages of energy gained from fat increased notably over time, exceeding the recommended 30% in 2006 and reaching 34.2% in 2018. In comparison, the amino acid components of soy protein are similar to the EAAs of humans, whereas it contains no cholesterol. However, our study showed that only around 10% of the total dietary protein intake came from legume sources from 1991 to 2018, which was less than 10 g/day for all groups. This was far below the recommended intake level of soy and soy products suggested by the Chinese Society of Nutrition, which is 30–50 g/day.

This transition in the source of dietary protein can be explained by the dietary Westernization in China. The traditional Chinese diet includes large amounts of cereals and vegetables and only small amounts of animal source foods. The Chinese modernization that occurred over the past 30 years has developed dramatically. The China National Nutrition Survey indicated that the population’s food consumption patterns have changed dramatically and experienced a shift from traditional dietary patterns to Western dietary patterns. Between 1991 and 2011, the consumption of animal source foods increased; over the same time period, the consumption of coarse grains, legumes, vegetables, and fruits remained low from 1982 to 2012 [16]. The current study indicates that the contribution from cereals was more than 50% from 1991 to 1993 and decreased to 48% in 1997. After the reform and opening up of China, the entire food system and ways of cooking and eating changed rapidly, including a decline in the proportion of food cooked in healthy ways (steamed, baked, or boiled) and an increase in the consumption of fried foods, snacking, and food consumption away from home. This occurred simultaneously with the very rapid modernization of the restaurant, packaged food manufacturing, and retail sectors [18]. Evidence shows that the intake of red meat has increased in the Chinese population over the past decades and that fatty fresh pork has been the predominant component of total meat consumed over time, which is particularly rich in saturated fatty acids and cholesterol [27]. Unlike in the United States and other high-income countries, where two-thirds of the protein intake comes from animal sources, in some Asian countries, like China and Korea, only one-third of the protein intake comes from animal sources among the elderly [23]. Although plant proteins are supposed to be incomplete, due to low digestibility and bioavailability, as well as insufficient amounts of EAAs, epidemiological studies have documented that plant source diets can lower the risk of obesity, diabetes, hypertension, and metabolic syndrome. In addition, researchers have suggested that consuming cereals and legumes together with higher amounts of plant-based products can enhance the nutritional quality. Furthermore, to minimize the adverse health and environmental effects of excess animal protein consumption, the incorporation of sustainably sourced plant proteins may be a promising strategy [20,28]. Hence, no definitive evidence is available regarding the possibly differing effects of animal-derived and planted-based proteins on muscle health in old age [14]. Future analyses should evaluate what amounts and which dietary protein sources may offer the greatest benefits for muscle health among elderly Chinese people.

The strengths of the study included using the most recent dietary data from the CHNS, investigating the trends of dietary protein intake levels and food sources over the past 28 years among the elderly Chinese population, and examining recent conformity with protein-specific DRIs. There were some limitations in our study. First, the method of using three consecutive days of 24 h dietary recalls posed the risk of underestimation of the dietary intakes for episodically consumed foods. Second, recall bias from self-reported measurements may exist. Third, we did not examine the use of protein supplements because supplement use was reported by only 1.75% of elderly Chinese individuals aged above 60 years. In addition, there is limited information available on the use of supplements in the CHNS.

## 5. Conclusions

This study provided important insights into the protein intake status transition among elderly Chinese individuals over a 28-year period. Elderly Chinese individuals experienced a significant decrease in their dietary protein intake and had a high prevalence of protein intake below the EAR (0.9 g/kg/day) and the RNI (1.0 g/kg/day). The transformation of their dietary patterns had positive effects on improving dietary protein quality, but it may also result in the development and exacerbation of adverse health conditions due to increases in animal source food consumption. Targeted interventions and education may be necessary to encourage the elderly Chinese population to meet the minimum protein intake recommendations by increasing legume consumption or using protein supplementation.

## Figures and Tables

**Figure 1 nutrients-13-03806-f001:**
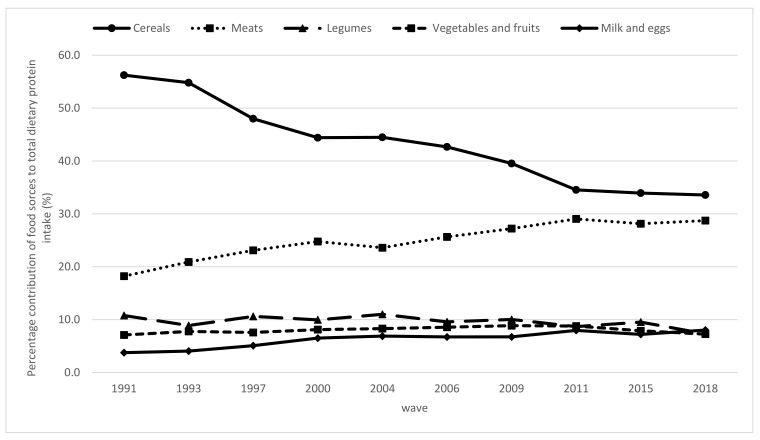
Trends in percentage contribution of food sources to total dietary protein intake among the elderly Chinese by wave from 1991 to 2018 (*p* < 0.05 for the trend for all).

**Figure 2 nutrients-13-03806-f002:**
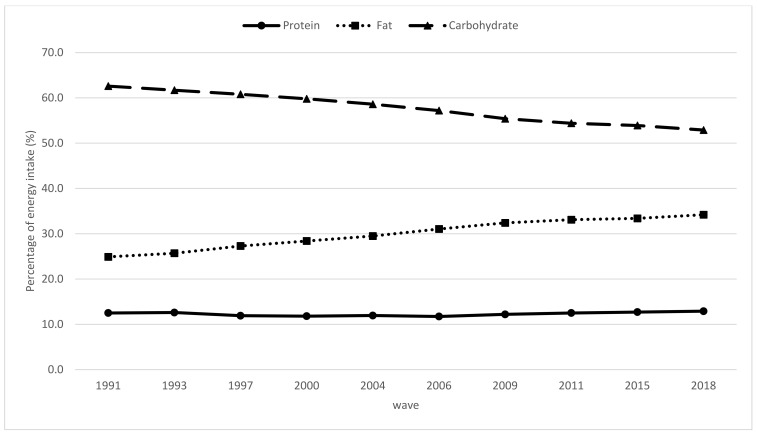
Trends in percentage of energy from protein, fats, and carbohydrates, among the elderly Chinese by wave from 1991 to 2018 (*p* < 0.05 for the trend for all).

**Table 1 nutrients-13-03806-t001:** Demographic characteristics of samples, 1991 to 2018 ^a^.

Wave	1991	1993	1997	2000	2004	2006	2009	2011	2015	2018	*p* Value ^b^
Sample size (*n*)	1334	1372	1588	1893	2144	2359	2662	3655	4994	5870	
Age group (years)											
60–69	855 (64.1)	890 (64.9)	999 (62.9)	1169 (61.8)	1245 (58.1)	1344 (57.0)	1523 (57.2)	2208 (60.4)	3180 (63.7)	3655 (62.3)	<0.0001
70–	479 (35.9)	482 (35.1)	589 (37.1)	724 (38.2)	899 (41.9)	1015 (43.0)	1139 (42.8)	1447 (39.6)	1814 (36.3)	2215 (37.7)	
Gender											
Male	633 (47.5)	646 (47.1)	732 (46.1)	883 (46.6)	1012 (47.2)	1100 (46.6)	1252 (47.0)	1733 (47.4)	2355 (47.2)	2748 (46.8)	0.99
Female	701 (52.5)	726 (52.9)	856 (53.9)	1010 (53.4)	1132 (52.8)	1259 (53.4)	1410 (53.0)	1922 (52.6)	2639 (52.8)	3122 (53.2)	
Education level											
Primary/illiterate	1176 (90.6)	1142 (88)	1235 (85.2)	1369 (79.7)	1604 (75.4)	1709 (73.2)	1905 (72.1)	2289 (62.9)	2798 (56.3)	2842 (49.9)	<0.0001
Middle school and above	122 (9.4)	155 (12)	215 (14.8)	348 (20.3)	524 (24.6)	625 (26.8)	738 (27.9)	1350 (37.1)	2173 (43.7)	2850 (50.1)	
Yearly income level											
Low	441 (33.3)	453 (33.3)	520 (33.3)	610 (33.3)	709 (33.4)	773 (33.3)	872 (33.3)	1202 (33.3)	1625 (33.3)	1723 (33.4)	1.00
Middle	443 (33.4)	454 (33.4)	520 (33.3)	611 (33.4)	707 (33.3)	776 (33.4)	872 (33.3)	1202 (33.3)	1624 (33.3)	1719 (33.3)	
High	442 (33.3)	453 (33.3)	520 (33.3)	610 (33.3)	709 (33.4)	773 (33.3)	872 (33.3)	1202 (33.3)	1625 (33.3)	1722 (33.3)	
Residence area											
City	366 (27.4)	343 (25)	414 (26.1)	461 (24.4)	548 (25.6)	605 (25.6)	652 (24.5)	1253 (34.3)	1895 (37.9)	2377 (40.5)	<0.0001
Suburb	453 (34)	483 (35.2)	589 (37.1)	620 (32.8)	649 (30.3)	717 (30.4)	835 (31.4)	915 (25)	1014 (20.3)	1165 (19.8)	
Town	420 (31.5)	431 (31.4)	495 (31.2)	530 (28)	583 (27.2)	631 (26.7)	710 (26.7)	982 (26.9)	1516 (30.4)	1677 (28.6)	
Village	95 (7.1)	115 (8.4)	90 (5.7)	282 (14.9)	364 (17)	406 (17.2)	465 (17.5)	505 (13.8)	569 (11.4)	651 (11.1)	

^a^ The values are expressed as numbers (percentages). ^b^ Chi-square tests are used to compare differences in categorical variables.

**Table 2 nutrients-13-03806-t002:** Trends in daily protein intake (g) among the elderly Chinese, 1991 to 2018 ^a^.

Wave	1991	1993	1997	2000	2004	2006	2009	2011	2015	2018	Per-Year Change (β ± SE)	*p* Value for Linear Trend ^b^
Sample size *(n*)	1334	1372	1588	1893	2144	2359	2662	3655	4994	5870		
Age group (years)												
60–69	66.1 (30.6)	64.3 (28.0)	63.2 (29.6)	59.8 (28.0)	62.6 (31.8)	61.6 (30.7)	61.1 (28.9)	56.0 (28.0)	58.8 (31.5)	59.8 (31.0)	−0.033 ± 0.0002	<0.0001
70–	57.4 (30.3)	57.0 (27.2)	54.1 (28.6)	54.2 (27.2)	54.6 (31.0)	53.1 (30.2)	51.9 (25.2)	50.3 (26.6)	52.2 (30.7)	54.6 (31.2)	−0.029 ± 0.0001	<0.0001
*p* value for interaction ^c^												0.03
Gender												
Male	68.5 (33.2)	67.0 (29.8)	65.4 (30.7)	62.9 (30.9)	65.0 (31.9)	62.7 (32.6)	63.3 (30.6)	58.6 (28.3)	60.9 (32.1)	62.8 (33.1)	−0.034 ± 0.0002	<0.0001
Female	58.3 (28.5)	57.9 (25.2)	56.1 (27.2)	54.0 (24.8)	54.9 (29.3)	54.2 (29.8)	52.8 (24.3)	49.7 (24.2)	51.8 (29.0)	53.9 (29.3)	−0.030 ± 0.0002	<0.0001
*p* value for interaction ^c^												0.88
Education level												
Primary/illiterate	62.5 (29.9)	61.3 (28.1)	58.7 (28.4)	55.0 (26.3)	57.6 (30.0)	55.1 (30.9)	54.8 (25.9)	50.5 (25.6)	51.6 (28.3)	53.3 (29.9)	−0.030 ± 0.0000	<0.0001
Middle school and above	72.3 (35.9)	71.1 (33.9)	71.1 (28.9)	68.2 (27.0)	68.1 (37.6)	64.7 (31.3)	63.9 (30.7)	59.8 (28.0)	62.6 (32.0)	62.8 (31.9)	−0.036 ± 0.0002	<0.0001
*p* value for interaction ^c^												0.04
Yearly income level												
Low	58.2 (30.7)	56.7 (27.5)	58.3 (31.5)	54.1 (27.8)	56.0 (29.8)	52.5 (32.4)	52.2 (25.7)	49.2 (25.4)	50.3 (27.9)	51.5 (29.4)	−0.029 ± 0.0001	<0.0001
Middle	60.5 (29.6)	61.4 (27.8)	58.6 (27.1)	55.8 (27.9)	59.0 (30.6)	57.5 (29.6)	58.7 (26.6)	53.7 (24.8)	56.3 (29.5)	57.5 (30.7)	−0.032 ± 0.0002	<0.0001
High	71.1 (30.5)	66.6 (28.5)	63.4 (28.9)	62.0 (28.3)	63.6 (32.3)	63.2 (31.5)	60.4 (29.7)	59.5 (29.4)	64.1 (33.0)	63.5 (31.6)	−0.035 ± 0.0002	<0.0001
*p* value for interaction ^c^												0.44
Area												
City	68.3 (31.5)	62.7 (25.7)	65.1 (28.8)	63.2 (31.3)	64.5 (34.2)	66.9 (31.1)	62.5 (33.3)	59.5 (31.1)	63.1 (32.9)	63.4 (33.5)	−0.035 ± 0.0002	<0.0001
Suburb	62.9 (31.5)	60.7 (30.3)	57.3 (30.1)	57.5 (26.9)	58.8 (32.1)	57.8 (31.6)	56.1 (25.3)	52.6 (25.9)	54.8 (28.8)	59.1 (29.5)	−0.031 ± 0.0001	<0.0001
Town	58.2 (25.2)	58.8 (27.3)	60.0 (28.0)	55.0 (24.6)	56.5 (27.0)	50.0 (23.9)	53.2 (26.7)	49.0 (22.9)	51.9 (28.0)	51.5 (29.0)	−0.029 ± 0.0001	<0.0001
Village	82.5 (34.2)	72.8 (40.0)	55.6 (24.8)	56.8 (27.3)	56.5 (31.0)	62.1 (36.1)	57.4 (24.3)	53.3 (26.7)	51.5 (28.8)	52.4 (29.9)	−0.033 ± 0.0003	<0.0001
*p* value for interaction ^c^												<0.0001
Total	63.3 (30.7)	61.5 (28.7)	60.0 (29.1)	57.5 (27.3)	59.5 (31.1)	57.8 (31.9)	57.2 (27.2)	53.7 (27.3)	56.5 (31.1)	57.8 (31.2)	−0.032 ± 0.0001	<0.0001

^a^ The values are expressed as medians (interquartile range). ^b^ Multivariable linear regression models include survey year as a continuous variable adjusted for all covariates. ^c^
*p* values for interaction between socioeconomic variables and trend variable in multivariable linear regression analyses adjusted for all covariates.

**Table 3 nutrients-13-03806-t003:** Trends in proportion of daily protein intake below EAR among the elderly Chinese, 1991 to 2018 ^a^.

Wave	1991	1993	1997	2000	2004	2006	2009	2011	2015	2018	Per-Year Change (β ± SE)	*p* Value for Linear Trend ^b^
Total subjects	475 (35.6)	478 (34.8)	633 (39.9)	810 (42.8)	879 (41.0)	1025 (43.5)	1176 (44.2)	1894 (51.8)	2362 (47.3)	2581 (44.0)	0.024 ± 0.0017	<0.0001
Age group (years)												
60–69	258 (30.2)	269 (30.2)	330 (33.0)	450 (38.5)	439 (35.3)	498 (37.1)	548 (36.0)	1028 (46.6)	1378 (43.3)	1473 (40.3)	0.027 ± 0.0022	<0.0001
70–	217 (45.3)	209 (43.4)	303 (51.4)	360 (49.7)	440 (48.9)	527 (51.9)	628 (55.1)	866 (59.8)	984 (54.2)	1108 (50.0)	0.021 ± 0.0027	<0.0001
*p* value for interaction ^c^												0.0033
Gender												
Male	225 (35.5)	246 (38.1)	301 (41.1)	395 (44.7)	428 (42.3)	500 (45.5)	561 (44.8)	922 (53.2)	1131 (48.0)	1243 (45.2)	0.022 ± 0.0024	<0.0001
Female	250 (35.7)	232 (32.0)	332 (38.8)	415 (41.1)	451 (39.8)	525 (41.7)	615 (43.6)	972 (50.6)	1231 (46.6)	1338 (42.9)	0.027 ± 0.0023	<0.0001
*p* value for interaction ^c^												0.4367
Education level												
Primary/illiterate	423 (36.0)	393 (34.4)	511 (41.4)	640 (46.7)	689 (43.0)	790 (46.2)	903 (47.4)	1311 (57.3)	1510 (54.0)	1431 (50.4)	0.025 ± 0.0019	<0.0001
Middle school and above	33 (27.0)	48 (31.0)	54 (25.1)	91 (26.1)	182 (34.7)	225 (36.0)	264 (35.8)	574 (42.5)	844 (38.8)	1047 (36.7)	0.024 ± 0.0036	<0.0001
*p* value for interaction ^c^												0.0054
Yearly income level												
Low	196 (44.4)	196 (43.3)	229 (44.0)	307 (50.3)	333 (47.0)	400 (51.7)	469 (53.8)	741 (61.6)	945 (58.2)	955 (55.4)	0.026 ± 0.0028	<0.0001
Middle	173 (39.1)	156 (34.4)	227 (43.7)	279 (45.7)	292 (41.3)	339 (43.7)	352 (40.4)	633 (52.7)	791 (48.7)	761 (44.3)	0.019 ± 0.0029	<0.0001
High	103 (23.3)	122 (26.9)	168 (32.3)	197 (32.3)	244 (34.4)	268 (34.7)	325 (37.3)	493 (41.0)	563 (34.6)	580 (33.7)	0.029 ± 0.0031	<0.0001
*p* value for interaction ^c^												0.9920
Area												
City	105 (28.7)	97 (28.3)	126 (30.4)	148 (32.1)	180 (32.8)	173 (28.6)	222 (34.0)	513 (40.9)	682 (36.0)	814 (34.2)	0.021 ± 0.0032	<0.0001
Suburb	172 (38.0)	193 (40.0)	263 (44.7)	268 (43.2)	264 (40.7)	323 (45.0)	381 (45.6)	496 (54.2)	505 (49.8)	477 (40.9)	0.014 ± 0.0030	<0.0001
Town	184 (43.8)	159 (36.9)	200 (40.4)	258 (48.7)	261 (44.8)	367 (58.2)	368 (51.8)	622 (63.3)	852 (56.2)	932 (55.6)	0.029 ± 0.0030	<0.0001
Village	14 (14.7)	29 (25.2)	44 (48.9)	136 (48.2)	174 (47.8)	162 (39.9)	205 (44.1)	263 (52.1)	323 (56.8)	358 (55.0)	0.051 ± 0.0055	<0.0001
*p* value for interaction ^c^												<0.0001

^a^ The values are expressed as numbers (percentages). ^b^ Multivariable logistic regression models include survey year as a continuous variable adjusted for all covariates. ^c^
*p* values for interaction between socioeconomic variables and trend variable in multivariable logistic regression analyses adjusted for all covariates.

**Table 4 nutrients-13-03806-t004:** Trends in proportion of daily protein intake above RNI among the elderly Chinese, 1991 to 2018 ^a^.

Wave	1991	1993	1997	2000	2004	2006	2009	2011	2015	2018	Per-Year Change (β ± SE)	*p* Value for Linear Trend ^b^
Total subjects	756 (56.7)	750 (54.7)	815 (51.3)	894 (47.2)	1068 (49.8)	1121 (47.5)	1221 (45.9)	1411 (38.6)	2203 (44.1)	2775 (47.3)	−0.023 ± 0.0017	<0.0001
Age group (years)												
60–69	536 (62.7)	530 (59.6)	573 (57.4)	603 (51.6)	685 (55.0)	715 (53.2)	809 (53.1)	952 (43.1)	1524 (47.9)	1849 (50.6)	−0.025 ± 0.0021	<0.0001
70–	220 (45.9)	220 (45.6)	242 (41.1)	291 (40.2)	383 (42.6)	406 (40.0)	412 (36.2)	459 (31.7)	679 (37.4)	926 (41.8)	−0.021 ± 0.0027	<0.0001
*p* value for interaction ^c^												0.0277
Gender												
Male	362 (57.2)	343 (53.1)	374 (51.1)	411 (46.5)	505 (49.9)	515 (46.8)	591 (47.2)	664 (38.3)	1024 (43.5)	1274 (46.4)	−0.023 ± 0.0024	<0.0001
Female	394 (56.2)	394 (56.2)	394 (56.2)	394 (56.2)	394 (56.2)	394 (56.2)	394 (56.2)	394 (56.2)	394 (56.2)	394 (56.2)	−0.025 ± 0.0022	<0.0001
*p* value for interaction ^c^												0.7966
Education level												
Primary/illiterate	662 (56.3)	629 (55.1)	615 (49.8)	591 (43.2)	768 (47.9)	760 (44.5)	814 (42.7)	764 (33.4)	1046 (37.4)	1162 (40.9)	−0.024 ± 0.0019	<0.0001
Middle school and above	81 (66.4)	94 (60.6)	142 (66.0)	229 (65.8)	294 (56.1)	349 (55.8)	402 (54.5)	642 (47.6)	1144 (52.6)	1558 (54.7)	−0.022 ± 0.0035	<0.0001
*p* value for interaction ^c^												0.0386
Yearly income level												
Low	212 (48.1)	207 (45.7)	250 (48.1)	243 (39.8)	314 (44.3)	311 (40.2)	323 (37.0)	361 (30.0)	544 (33.5)	632 (36.7)	−0.024 ± 0.0028	<0.0001
Middle	230 (51.9)	249 (54.8)	246 (47.3)	271 (44.4)	348 (49.2)	357 (46.0)	430 (49.3)	439 (36.5)	694 (42.7)	800 (46.5)	−0.018 ± 0.0028	<0.0001
High	309 (69.9)	288 (63.6)	306 (58.8)	348 (57.0)	398 (56.1)	435 (56.3)	455 (52.2)	591 (49.2)	919 (56.6)	990 (57.5)	−0.027 ± 0.0030	<0.0001
*p* value for interaction ^c^												0.8876
Area												
City	238 (65.0)	206 (60.1)	256 (61.8)	270 (58.6)	322 (58.8)	370 (61.2)	367 (56.3)	621 (49.6)	1035 (54.6)	1345 (56.6)	−0.022 ± 0.0031	<0.0001
Suburb	254 (56.1)	245 (50.7)	278 (47.2)	282 (45.5)	326 (50.2)	340 (47.4)	379 (45.4)	331 (36.2)	421 (41.5)	585 (50.2)	−0.016 ± 0.0030	<0.0001
Town	188 (44.8)	216 (50.1)	247 (49.9)	216 (40.8)	257 (44.1)	193 (30.6)	269 (37.9)	268 (27.3)	542 (35.8)	602 (35.9)	−0.027 ± 0.0030	<0.0001
Village	76 (80.0)	83 (72.2)	34 (37.8)	126 (44.7)	163 (44.8)	218 (53.7)	206 (44.3)	191 (37.8)	205 (36.0)	243 (37.3)	−0.053 ± 0.0055	<0.0001
*p* value for interaction ^c^												<0.0001

^a^ The values are expressed as numbers (percentages). ^b^ Multivariable logistic regression models include survey year as a continuous variable adjusted for all covariates. ^c^
*p* values for interaction between socioeconomic variables and trend variable in multivariable logistic regression analyses adjusted for all covariates.

## Data Availability

Data sharing is not applicable to this article.

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
