# Peer review of "Dietary Protein Intake Dynamics in Elderly Chinese from 1991 to 2018"

_nutrients, 2021, doi:10.3390/nu13113806_

Round 1

Reviewer 1 Report

There ware several issues that need to be revised in this manuscript.

Major points

・Line 92:Please specify whether the food survey was conducted on three consecutive days (72h recall?).

・Table 2:Please consider the reasons why protein intake in "village" in 1991 and 1992 appears to be extremely high.

・Line 187:Please add the results for protein intake per kg of body weight. For example, if it is less than 0.8 g/kg, an absolute intake deficiency may be a concern.

Minor points

・Line 205:Please add the reference.

Reviewer 2 Report

In my opinion, the manuscript was well prepared.

  • The Introduction Section explains the design of the study. The Authors well justify the research topic.
  • The study was carried out without methodological errors.
  • The Descriptions of the results were correct.
  • The presented figures and table were prepared precisely and also legible.
  • The Discussion Section includes the accurate reference of the results obtained to the studies of other studies. 
  • The Conclusions were well formulated.

Reviewer 3 Report

The paper of Ouyang et al. is very interesting. They used simple but effective analysis to show the trend of dietary protein intake in elderly Chinese and they were careful to their limitations and explained easily the causes of this trend.

Minor revision

Please report also in the introduction the extension (EAR -RNI)

In lines, 102-103 report also where this formula was taken, as a reference or a justification

Please check table 3 and 4 there is no the IQR

Reviewer 4 Report

The authors conduct a longitudinal household-based study(similar a cross‐sectional study?) and aimed to investigate the dietary protein intake dynamics (is it dynamics?) according to economic, sociological, and demographic transformations in elderly Chinese from 1991 2 to 2018.

Comments:

1.

Table 2

P value for linear trend (multivariable linear regression models include survey year as a continuous variable adjusted for all covariates) should be provided beta coefficient (β, 95% CI or SE) and R squared (coefficient of determination), testing a per categories by year increasing or decreasing.  

2.

Table 2, Table 3, and Table 4 are similar.

P value for linear trend should be provided beta coefficient (β, 95% CI or SE; odds ratio 95% CI) and R squared or pseudo-R squared.

3.

Similarly, Figure 1 and Figure 2 should also be provided beta coefficient (β, 95% CI or SE)) and p vales.

4.

The statistical results by using multiple linear regression model or multiple logistic regression models should be reported in the abstract, e.g., β, 95% CI or SE, or odds ratio 95% CI and p-value.

5.

How to explained the significant interaction term of variables by survey year in Tables 2, 3 and 4(p for interaction <0.05) in this study? p for interaction <0.05, Why?

Round 2

Reviewer 4 Report

No further comment